

# Lactylation's role in bone health and disease: mechanistic insights and therapeutic potential

Zhiyuan Ye[1], Xuanyu Chen[1], Jiyuan Zou[1], Wenjing Wu[1], Jingyuan Yang[1], Li Yang[1,2], Lvhua Guo[1] and Tao Luo[1]

[1] Department of Stomatology, Guangzhou Medical University, Guangzhou, Guangdong Province, China
[2] Department of Endodontics, Guangzhou Medical University, Guangzhou, China

Corresponding authors
Lvhua Guo,
2010686002@gzhmu.edu.cn
Tao Luo, 2011686028@gzhmu.edu.cn

## ABSTRACT

This article reviews the mechanisms and research progress of lactylation, an emerging post-translational modification (PTM) of proteins, in bone metabolism and related diseases. Lactate-derived lactylation modifies lysine residues on histones, affecting chromatin structure and gene expression. Studies indicate that lactylation plays a significant role in bone metabolism, with mechanisms including the regulation of osteoblast differentiation, potential influence on osteoclast activity, and indirect effects on bone homeostasis through the modulation of immune cell functions such as macrophages and T cells. In periodontitis, lactylation may impact inflammation progression and tissue repair by regulating macrophage polarization and function. In osteoporosis, lactylation adjusts bone density by influencing osteogenic gene expression. Additionally, the role of lactylation in other skeletal behaviors and diseases is gradually being revealed, such as its association with insulin resistance in skeletal muscle, and its roles in tooth development and rheumatoid arthritis, providing new targets for the treatment of these conditions. Future research will focus on the enzymatic regulatory mechanisms of lactylation, its interactions with other PTMs, and its involvement in metabolic diseases and inflammatory responses. The regulation of lactylation offers new strategies for the treatment of bone-related diseases, including the development of drugs that can reverse or modulate lactylation, and the restoration of bone metabolic balance through the adjustment of lactylation levels. As the understanding of lactylation's regulatory mechanisms and biological functions deepens, its potential for clinical applications will continue to expand, particularly in the fields of bone regeneration, immunity, and the treatment of metabolic diseases.

## INTRODUCTION

Lactylation, a novel post-translational modification (PTM) of proteins, was first identified by *Zhang et al. (2019)*. This modification involves the addition of lactate-derived groups to lysine residues on histones, occurring both intracellularly and extracellularly, thereby impacting chromatin structure and gene expression. Lactate production is closely associated with glycolysis, glucose, and glycogen metabolism under fully aerobic conditions, as well as during high-intensity physical activity (*Brooks, 2018*). Historically considered a waste
product of energy metabolism, lactate has been redefined as a crucial metabolic intermediate with key roles in energy metabolism and signaling—a concept known as the "lactate shuttle" (*Brooks, 2018*; *Brooks, 2020*). Currently, lactate serves as the precursor for lactylation, a new type of post-translational modification (PTM). As lactate accumulates, lactylation levels increase, regulating cellular metabolic processes (*Zhang et al., 2019*). Unlike other PTMs such as acetylation and methylation, lactylation underscores a novel role for lactate, a metabolic byproduct, in epigenetic regulation (*Yu et al., 2024*; *Zhang et al., 2019*).

In recent years, research into lactylation within biological systems has intensified, uncovering its connections to various pathophysiological processes, including tumorigenesis and inflammatory responses (*Xie et al., 2022*). Here, a brief overview is provided.

As early as the 20th century, the introduction of the Warburg effect shifted the understanding of energy metabolism within tumors. It demonstrated that even under aerobic conditions, the tumor microenvironment favors glycolysis for energy production, leading to the generation of lactate. Within tumors, lactate can be shuttled between different cells, supporting both the energy metabolism and biosynthetic needs of tumor cells, optimizing energy utilization. Today, lactate-derived lactylation participates in the epigenetic regulation of tumor cells, influencing gene expression and cellular function (*Wang et al., 2023*; *Xie et al., 2022*). Moreover, elevated lactylation levels within tumors positively affect tumor progression and metastasis. Specifically, lactylation reshapes the tumor microenvironment by altering inflammatory states and inducing angiogenesis, thereby promoting tumor growth (*Wang et al., 2023*). Lactylation-mediated immune suppression and immune evasion in tumors also provide new perspectives for targeted immunotherapy (*Su et al., 2023*).

The role of lactylation in inflammatory responses is complex and multifaceted. From a systemic perspective, sepsis, which leads to high mortality rates, is influenced by lactylation, acting both as an "accelerator" of the disease mechanism and as a "controller" of inflammation-related processes during treatment (*Liu et al., 2024a*). Current research is primarily focused on macrophages, where lactylation can affect inflammation through various pathways. For instance, lactate can act as a signaling molecule, influencing the activation of immune cells and the production of inflammatory factors *via* specific receptors like GPR81. On the other hand, lactylation can modify the function of inflammation-related proteins, regulating inflammatory processes and promoting the conversion of macrophages to the M2 type, aiding in tissue repair (*Gong et al., 2024*; *Hu et al., 2024*; *Wang et al., 2022a*; *Wei et al., 2024*).

While research on lactylation in cancer and inflammatory responses is relatively advanced, studies examining its effects in bone tissue—a tissue closely linked to lactate production—are limited. It is established that lactate accumulation influences lactylation levels (*Zhang et al., 2019*). The bone microenvironment is characterized by relatively low oxygen availability (*Marenzana & Arnett, 2013*), and daily physical activity, particularly intense exercise, increases lactate production in muscles and blood (*Billat et al., 2003*). Consequently, lactate levels in the bone microenvironment also rise (*Zhu et al., 2023*). This

raises the question: Does lactylation play a role in bone metabolism during these processes? This question merits further exploration.

## BONE METABOLISM

Bone metabolism is a complex and dynamic process that involves the formation and degradation of bone tissue to maintain structural integrity and mineral balance. At the cellular and molecular level, the interactions between osteoblasts, osteoclasts, and immune cells are crucial for maintaining this process (*Lorenzo, Horowitz & Choi, 2008*; *Ono et al., 2020*). Osteoblasts originate from multipotent mesenchymal stem cells (MSCs) and skeletal stem cells (SSCs) (*Salhotra et al., 2020*), and through a series of transcriptional regulatory steps, such as RUNX2 and Osterix, differentiate into mature osteoblasts (*Choi et al., 2024*). These cells secrete key factors that regulate bone formation, including bone morphogenetic proteins (BMPs), fibroblast growth factors (FGFs), and Wnt signaling pathway-related factors, which are involved in the synthesis and mineralization of the bone matrix, leading to new bone formation (*Shahi, Peymani & Sahmani, 2017*). In contrast, osteoclasts are responsible for bone resorption. Their differentiation, activation, and survival are critically regulated by the binding of Receptor Activator of Nuclear Factor-κB Ligand (RANKL) to its receptor RANK on osteoclast precursors (*Theoleyre et al., 2004*), and the RANK-RANKL axis plays a pivotal role in bone resorption (*Kim et al., 2020*). Immune cells, such as macrophages and T cells, act more like collaborators by expressing RANKL to promote osteoclastogenesis and activity, contributing to bone destruction in certain inflammatory diseases (*Ono et al., 2020*). The interaction of these cells plays an essential role in maintaining bone metabolic homeostasis. Bone homeostasis is a balanced process involving the coordinated actions of bone formation and resorption. Under normal conditions, bone formation and resorption are in dynamic equilibrium, ensuring the structural and functional integrity of bone tissue (*Lerner, Kindstedt & Lundberg, 2019*; *Yang & Liu, 2021*).

Recent research has indicated that lactylation plays an important role in the above-mentioned bone metabolism process. First, changes in lactylation levels can affect gene expression and regulate cellular metabolic functions. For example, lactylation is involved in regulating stem cell differentiation. Studies have shown that lactylation promotes osteoblast differentiation, playing a significant role in bone formation (*Wu et al., 2023a*). In macrophages, lactylation can activate the expression of anti-inflammatory genes, mediating inflammatory responses and tissue repair (*Chen et al., 2021a*), which highlights its key role in immune responses (*Pan et al., 2022*; *Zhang et al., 2019*). Furthermore, lactylation *in vivo* is closely associated with the onset and progression of various bone-related diseases, such as periodontitis, osteoporosis, and osteoarthritis (*Wang et al., 2024b*; *Wu et al., 2023a*; *Wu et al., 2024*).

In addition, bone metabolism is regulated by various endogenous and exogenous factors, including hormones, local factors, nutritional status, and mechanical stress. Hormones such as parathyroid hormone (PTH), calcitonin, sex hormones, and vitamin D play positive roles in regulating bone formation (*Bikle, 2012*; *Chen et al., 2021b*; *Felsenfeld &*

*Levine, 2015*; *Venken et al., 2008*). Local factors include but are not limited to RANKL, osteoprotegerin (OPG), BMPs, FGFs, and Wnt signaling molecules. These factors not only influence the differentiation and function of bone cells but also affect the morphology and quality of bone (*Shahi, Peymani & Sahmani, 2017*). Moreover, bone tissue is highly sensitive to mechanical stress. Physical stress can impact bone formation and resorption by regulating the activity of osteoblasts and osteoclasts. Adequate mechanical stimuli, such as exercise, can promote bone formation (*Huang et al., 2023*).

When the regulation of bone metabolism is disrupted, diseases such as osteoporosis may arise. Therefore, understanding the regulatory mechanisms of bone metabolism is crucial for the prevention and treatment of bone-related diseases (*Wawrzyniak & Balawender, 2022*). The discovery of lactylation has greatly expanded our understanding of the role of metabolic products in cellular regulation. Lactate, not only a byproduct of energy metabolism, also directly participates in gene expression regulation through PTM (*Zhang et al., 2019*). As research deepens, the role of lactylation in development, growth, and bone-related diseases will be further elucidated, and it may become an important therapeutic target in the future (*Dai et al., 2022*; *Wu et al., 2023a*). The following section will discuss in detail the impact of lactylation on bone metabolism.

## THE ROLE OF LACTYLATION IN BONE METABOLISM

### Lactylation regulates osteogenic differentiation

Stem cell differentiation is an energy-intensive process, with the energy sources relying on pathways such as aerobic glycolysis and oxidative phosphorylation (*Ning et al., 2022*). For example, self-renewing hematopoietic stem cells (HSCs) primarily rely on anaerobic glycolysis to maintain their quiescent state. However, during HSC differentiation, cells shift toward engagement of the tricarboxylic acid (TCA) cycle and oxidative phosphorylation (OXPHOS) to generate high levels of ATP, reflecting increased energy demands (*Casati et al., 2011*). Similarly, differentiated osteoclasts secrete large amounts of acids and proteolytic enzymes and require substantial energy, relying predominantly on OXPHOS as the main bioenergetic mechanism. In differentiating osteoblasts, a distinct metabolic pattern is observed: these cells are characterized by the presence of mitochondria with high transmembrane potential and a sharp increase in oxygen consumption (*Arnett & Orriss, 2018*). Nevertheless, even under aerobic conditions, osteoblasts primarily metabolize glucose through glycolysis into pyruvate. Lactate is generated from pyruvate through the glycolytic pathway, particularly under hypoxic conditions or during metabolic states characterized by rapid energy demands, where glycolysis produces ATP more quickly than oxidative phosphorylation (*Guntur et al., 2018*; *Shen, Hu & Karner, 2022*). These metabolic shifts, especially during stem cell differentiation and bone cell lineage commitment, are associated with changes in lactate production Lactate, previously considered a metabolic waste product, has recently been found to play a key role in energy metabolism (*Choi et al., 2024*; *Loeffler et al., 2018*; *Neto et al., 2023*), particularly in the osteogenic differentiation of stem cells. Lactylation has been shown to be closely associated with cellular energy status and metabolic activity (*Li et al., 2024b*), excessive lactate may contribute to the dynamic regulation of histone lactylation.

For instance, lactate-derived lactylation on histone lysine residues promotes gene transcription, increasing the expression of osteogenic differentiation-related transcription factors such as Runx2, JunB, and other osteogenesis-associated genes (*Li et al., 2025*), which in turn promotes the differentiation of stem cells into osteoblasts (*Nian et al., 2022*; *Wu et al., 2023a*). It is well known that lactate dehydrogenase A (LDHA) converts pyruvate into lactate, and within the cell, the levels of LDHA are closely associated with both the osteogenic differentiation and histone lactylation levels. Histone H3 lysine 18 (H3K18) is a key site for lactylation, and its marking indicates the activation of gene expression (*Wang et al., 2024a*). During osteogenic differentiation in various cell types, the increase in LDHA levels enhances both osteogenic differentiation and histone lactylation levels. Conversely, when LDHA is inhibited, both levels decrease. Chromatin immunoprecipitation (ChIP) assays have shown that LDHA activates osteogenic gene expression by influencing H3K18 lactylation (H3K18la) (*Minami et al., 2023*; *Nian et al., 2022*) (Fig. 1).

The terms "Writer" and "Eraser" refer to enzymes or proteins responsible for adding, promoting, or suppressing lactylation modifications. Among the "Writers", the role of histone acetyltransferase (p300) is particularly notable. For example, in the C2C12 myoblast cell line, the introduction of Ep300 siRNA into C2C12 cells suppressed the expression of Ep300 mRNA, lactylation levels, and osteogenesis-related gene expression. However, during this process, both intracellular and extracellular lactate concentrations remained unaffected, indicating that p300 is a key factor controlling lactylation levels (*Minami et al., 2023*). For the "Erasers", compounds such as the p300 inhibitor A485 can inhibit UDP-glucose dehydrogenase (UGDH) lactylation *in vitro* and *in vivo*, rescuing chondrocyte extracellular matrix degradation and delaying the progression of osteoarthritis (OA). Furthermore, pyruvate dehydrogenase kinase inhibitors (DCA) and histone deacetylases (HDACs) have been found to suppress lactylation levels, although their impact on osteogenic differentiation remains uncertain (*Moreno-Yruela et al., 2022*; *Wang et al., 2022b*; *Zhang et al., 2019*).

Existing evidence suggests that p300 is a relatively clear "Writer" of lactylation modifications. Its mechanism likely involves the interaction between p300 and GTP-specific succinyl-CoA synthetase (GTPSCS), forming a functional lactyltransferase complex *in vivo*, which promotes histone lactylation (*Liu et al., 2025*; *Zong et al., 2025*). Lysine acetyltransferases (KAT) and alanylal-tRNA synthetase (AARS) also exhibit "Writer" roles, but their research in the bone biology field is limited (*Ren, Tang & Zhang, 2025*). As for the "Erasers", compounds such as A485 inhibit p300, thereby indirectly suppressing lactylation levels rather than acting directly. DCA has yet to be shown to suppress osteogenic differentiation, and both A485 and DCA are small molecules, not enzymes or proteins. Therefore, HDACs are currently considered the most defined "Erasers" (*Ren, Tang & Zhang, 2025*). Additionally, SIRT2 is a newer "Eraser", but its catalytic efficiency is lower than that of HDACs (*Zong et al., 2025*). In conclusion, further exploration of "Erasers" in the osteogenic field is needed (*Cui et al., 2021*; *Hu et al., 2024*; *Minami et al., 2023*; *Yang et al., 2022*; *Zhang et al., 2019*).

Currently, no specific bone-specific regulators have been identified to regulate the lactylation response. Lactylation levels are primarily controlled by lactate levels and

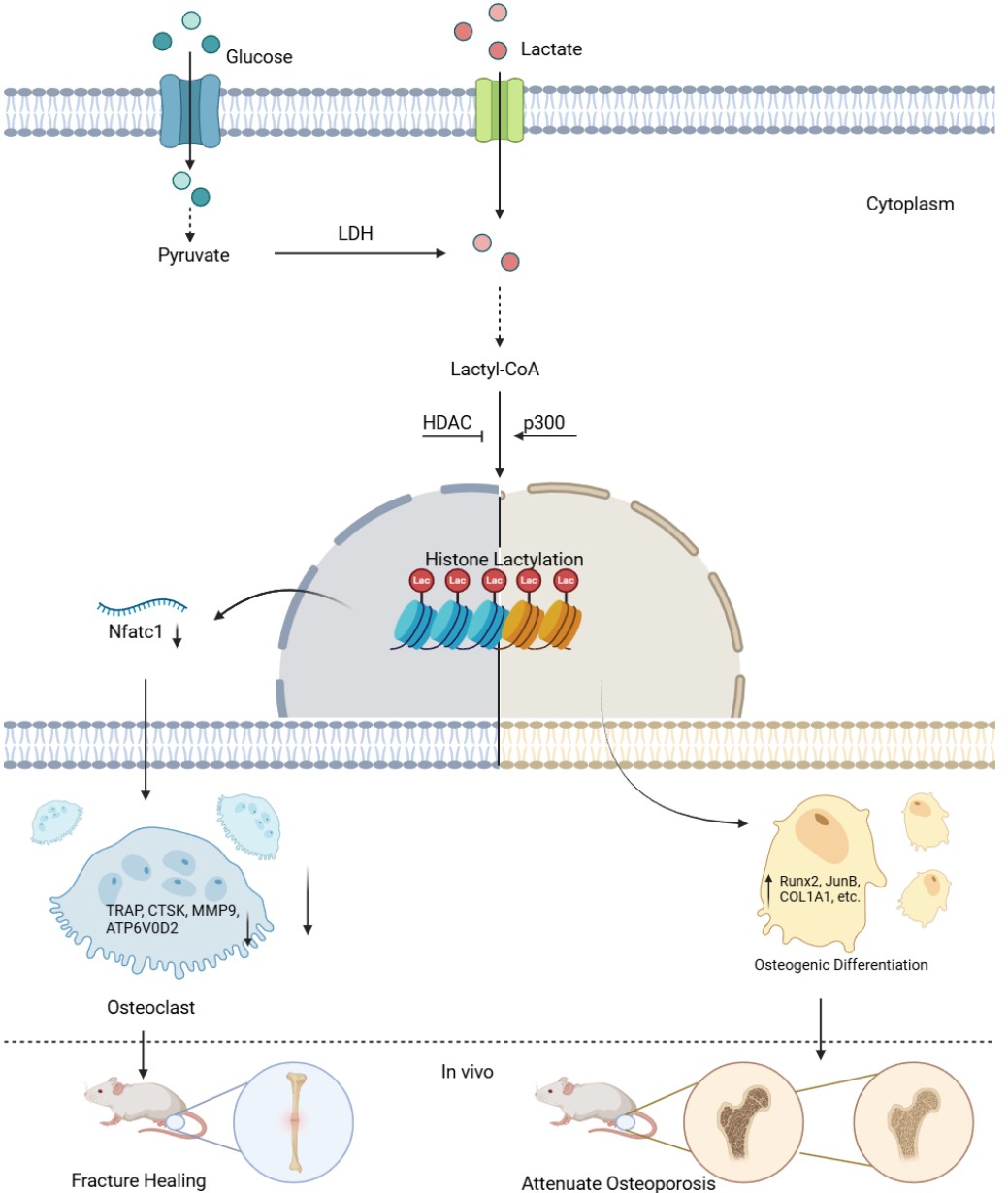

**Figure 1 Lactylation in osteoblasts and osteoclasts.** Lactate derived from both intracellular and extracellular sources is converted to lactyl-CoA, which serves as the substrate for histone lactylation. In the nucleus, "Writer" enzymes (primarily p300) catalyze the addition of lactyl groups to lysine residues on histones. This modification is reversible through the action of "Eraser" enzymes (notably HDACs). The lactylation modification regulates gene transcription in a cell-type specific manner: In osteoclast precursors, it suppresses osteoclastogenesis by downregulating *Nfatc1* expression and subsequent osteoclast-specific genes (*TRAP*, *CTSK*, *MMP9*, *ATP6V0D2*). In stem cells, it promotes osteogenic differentiation by enhancing transcription of osteogenic markers (*Runx2*, *JunB*, *COL1A1*), thereby facilitating bone formation *in vivo*.

enzymes that promote lactylation modifications, with lactate levels being mainly influenced by factors such as LDH and pyruvate kinase (PK) (*Brooks et al., 1999*; *Llanos et al., 1993*).

## Lactation and osteoclasts

While emerging studies have explored the role of lactylation in regulating osteoblast function, direct evidence for the regulation of osteoclasts by lactylation remains scarce. Nevertheless, this is an intriguing and promising area of investigation. Recent studies indicate that RANKL induces a metabolic shift in osteoclasts toward accelerated glycolytic metabolism (*Taubmann et al., 2020*), and inhibition of glycolysis has been shown to ameliorate osteoclast activity and bone resorption. Additionally, a recent study by *Shen et al. (2025)* reported that $Mg^{2+}$ promotes lactate production in osteoclasts, leading to increased histone lactylation and suppression of osteoclastogenesis (Fig. 1).

Although direct studies specifically addressing lactylation in osteoclasts are currently limited, interest in the role of lactylation in bone biology is rapidly expanding. Moreover, regulation of osteoclast function through other PTMs provides structural and mechanistic support for the potential importance of lactylation. For example, *Williams, Nishu & Rahman (2011)* demonstrated that inhibition of HDACs, known erasers of lactylation and acetylation, suppresses osteoclastogenesis by upregulating the expression of C/EBP-β and MKP-1. Similarly, *Kim et al. (2011)* showed that RANKL induces NFATc1 acetylation *via* histone acetyltransferases (HATs), enhancing NFATc1 stability and transcriptional activity during osteoclast differentiation. These findings suggest that metabolism-sensitive PTMs, including lactylation, may play critical regulatory roles in osteoclast biology and highlight important directions for future research.

Lactate plays a complex role in bone metabolism, particularly in the differentiation, activity, and function of osteoclasts. In an acidic environment, the activity of osteoclasts is enhanced, and the accumulation of lactate can lower the local pH value, which may promote the fusion, differentiation, activity, maturation, and function of osteoclasts (*Arnett & Orriss, 2018*; *Yuan et al., 2016*). Interestingly, in osteoclasts, lactate accumulation and the resulting decrease in pH have opposite effects compared to the lactylation induced by lactate accumulation, leading to conflicting results. Therefore, the relationship between lactylation and osteoclasts remains to be further clarified.

## Lactylation and immune cell regulation

Bone homeostasis is a part of bone metabolism, where the roles of osteoblasts and osteoclasts are self-evident (*Shahi, Peymani & Sahmani, 2017*). However, the regulation of bone homeostasis is also inseparable from the function of immune cells, among which macrophages are closely related to bone homeostasis (*Bozec & Soulat, 2017*; *Gu, Yang & Shi, 2017*; *Hu et al., 2023*; *Hu et al., 2022*). For instance, regarding osteoblasts, M2-type macrophages can secrete osteogenic inducers such as BMP-2, thereby stimulating the expression of osteogenic gene phenotypes in osteoblasts, such as Runx2, ALP, and OCN, affecting osteoblast differentiation and osteogenic activity. In contrast, M2-type macrophages can also secrete anti-inflammatory factors like IL-4 and IL-13, with IL-4 capable of preventing osteoclast formation by activating the NF-κB pathway. In opposition

to M2-type, M1-type macrophages promote the generation of osteoclasts by secreting pro-inflammatory factors such as TNF-α and IL-1β, and inhibit the expression of osteogenic factors like Runx2 and IGF-1, thereby suppressing osteoblast production. M1-polarized macrophages can also mediate the RANKL/RANK system, causing osteoclast-induced bone damage (*Bozec & Soulat, 2017*; *Hu et al., 2023*; *Hu et al., 2022*). Thus, it is evident that macrophages can influence bone homeostasis by affecting the metabolic states of osteoblasts and osteoclasts.

Recent research has found that lactylation is an important regulator of macrophages, on the one hand, lactylation can regulate the phenotype of macrophages, and on the other hand, it can also regulate the metabolism of macrophages (*Xu et al., 2024*). Firstly, as a metabolic product, lactate accumulates in an acidic environment and can alter the phenotype of macrophages (*Wei et al., 2024*). Studies have shown that through glycolysis and the production of lactate, the B-cell adapter for PI3K (BCAP) promotes histone lactylation, which facilitates the transition of inflammatory macrophages to M2-type macrophages, beneficial for alleviating intestinal inflammation (*Irizarry-Caro et al., 2020*). Additionally, in a mouse model of colon cancer, lactate can also promote the M2 polarization of macrophages through the lactylation of retinoic acid-inducible gene 1 (RIG-1) (*Gu et al., 2024*). These transformations reveal a macro-level shift in the balance between the two types of macrophages within inflammatory models. Intracellularly, increased mitochondrial fission also leads to changes in lactylation levels, thereby inducing M2 macrophage polarization (*Susser et al., 2023*). The shift towards the M2 phenotype enables macrophages to more effectively promote anti-inflammatory responses and reparative functions (*Liu et al., 2019*; *Schlundt et al., 2021*).

Moreover, the production of lactate is closely associated with local hypoxic conditions, which not only promote lactate generation but also induce macrophages to produce more M2-type cells (*Riboldi et al., 2013*; *Zhou et al., 2022*). Studies have found that in hypoxic environments, macrophages can regulate the expression of osteoblast-related factors through HIF-1α (*Dehne & Brüne, 2009*), such as BMP-2 and VEGF. These factors not only directly participate in the osteogenic process but also promote angiogenesis, providing essential nutrients and oxygen to the bone (*Chen et al., 2012*; *Wang et al., 2020*). Therefore, the roles of hypoxia and lactate play a crucial dual role in maintaining bone homeostasis (*Karner & Long, 2018*). Further analysis of the metabolic effects of lactylation on macrophages reveals that lactylation has an important role in regulating gene expression in macrophages. H3K18la drives M2-like gene expression, such as Arg1, during M1 macrophage polarization, and this induction process is time-dependent (*Zhang et al., 2019*). Arg1, as a negative regulator of osteoclasts, can inhibit osteoclast differentiation (*Yeon, Choi & Kim, 2016*). In another study, tumor-derived lactate drove macrophages to downregulate RARγ gene transcription through histone lactylation, thereby promoting tumor growth (*Li et al., 2024a*), and RARγ also plays an important role in promoting osteoblast differentiation (*Green et al., 2017*). Although this was observed in a tumor model, the ability of lactylation to affect RARγ expression provides a new possibility for future research on the impact of lactylation on bone metabolism. Additionally, in a model of infected macrophages, the removal of lactylation at the histone H4K8 site through Sirtuin

2 (SIRT2) reduced the chemotactic ability of macrophages. Therefore, targeting SIRT2 and H4K8la modifications may help control macrophage-mediated inflammation (*Wenting et al., 2023*). In summary, lactylation affects macrophage phenotypes and metabolism through multiple pathways, thereby creating a benign repair microenvironment. Although this regulation is not within the bone microenvironment, it demonstrates the great potential of lactylation in maintaining bone homeostasis.

In addition to macrophages, T cells also play a pivotal role in maintaining bone homeostasis (*Li, Pi & Li, 2021*), influencing the functions and metabolism of osteoblasts and osteoclasts (*Fischer et al., 2019*). Among different T cell subsets, such as CD4+T cells, particularly Th17 cells, promote osteoclastogenesis by secreting factors like IL-17 and RANKL. Binding of RANKL to its receptor RANK stimulates the differentiation of osteoclast precursors into mature osteoclasts, thereby enhancing bone resorption. RANKL is a crucial factor for osteoclast formation, while OPG acts as its antagonist (*Takegahara, Kim & Choi, 2022*). The presence of T cells can promote the expression of OPG, inhibit the action of RANKL, and reduce osteoclast activity (*Zhang et al., 2020*). Furthermore, it is well-established that T-regulatory cells (Tregs) promote the proliferation and differentiation of osteoblasts by secreting anti-inflammatory cytokines such as TGF-β and IL-10, thereby facilitating bone formation and contributing to the maintenance of bone density and quality (*Huang et al., 2022*; *Zhu et al., 2020*). Additionally, T cells can regulate the production of RANKL and OPG by influencing the function of dendritic cells and other immune cells, thereby affecting osteoclastogenesis (*Walsh & Choi, 2014*).

Lactylation has been found to regulate the function of Treg cells. For instance, in the tumor microenvironment, lactylation can enhance TGF-β signaling by modulating specific proteins in Treg cells, such as MOESIN, thereby influencing the differentiation, stability, and function of Treg cells (*Gu et al., 2022*). Furthermore, lactylation not only affects the function of Treg cells but may also impact immune homeostasis by regulating the metabolic states of other T-cell subsets (*Chen et al., 2022b*; *Wu, Huang & Zhao, 2023*). Such effects hold significant regulatory importance for the bone marrow microenvironment, bone metabolism, and bone homeostasis (*Fischer et al., 2019*).

In the experimental autoimmune uveitis (EAU) mouse model, the lactylation level in CD4+ T cells significantly increases. The use of glycolysis inhibitors, such as dichloroacetic acid (DCA), a pyruvate dehydrogenase kinase inhibitor, can reduce the lactylation level in CD4+ T cells and inhibit the differentiation of TH17 cells. Conversely, blocking mitochondrial metabolism, for example with rotenone, increases the lactylation level in CD4+ T cells and promotes the differentiation of TH17 cells, alleviating the inflammatory process of EAU. Lactylation can influence the expression of specific genes in T cells by differentially regulating the binding of lactylated proteins at the Lys164 site to the promoters of TH17-related genes, such as Tlr4 and Runx1 (*Fan et al., 2023*), which are crucial for regulating bone metabolism and homeostasis (*Tang et al., 2021*; *Tominari et al., 2024*). In another study, lactate has a significant impact on T cells, particularly pro-inflammatory Th17 cells. Lactate reprogramming of Th17 cell metabolism and epigenetic status by inhibiting IL-17A production, inducing Foxp3 expression, and promoting histone H3K18la results in the transformation of these cells towards an immunosuppressive Tregs

phenotype, exhibiting an anti-inflammatory effect in intestinal inflammation models (*Lopez Krol et al., 2022*). For CD8 T cells, lactylation significantly affects their metabolic status. In activated CD8 T cells, lactate produced through glycolysis increases, whereas in memory CD8 T cells, lactate produced through mitochondrial metabolic pathways such as oxidative phosphorylation and fatty acid oxidation increases. Lactylation also regulates the function of CD8 T cells by affecting the expression of specific genes. For instance, H3K18la and H3K9la are enriched in the promoter regions of effector CD8 T cell genes such as Gzmb and IFN-γ, and positively correlate with the expression levels of these genes (*Raychaudhuri et al., 2024*). Gzmb and IFN-γ exhibit potent immune functions against the bone marrow microenvironment or intraosseous tumors (*Chen et al., 2022a*), while IFN-γ also contributes to osteoblast differentiation and chondrogenesis (*Maruhashi et al., 2015*) (Table 1).

Therefore, understanding the impact of lactylation on macrophages and various T-cell subsets will provide us with a new perspective to deepen our comprehension of their complex roles in bone metabolism and homeostasis. Although the regulatory mechanisms of lactylation on macrophages and T-cells have been partially elucidated, further research is required to fully understand how lactylation directly or indirectly affects bone homeostasis through macrophages and T-cells. However, it can be speculated that lactylation may influence the interaction between these immune cells and bone cells by modulating the function and activity of macrophages and T-cells (*Chen et al., 2021a*), ultimately impacting bone homeostasis.

## LACTYLATION IN BONE-RELATED DISEASES

### Lactylation and periodontitis

Periodontitis is an inflammatory disease caused by bacterial infection, typically accompanied by redness, swelling, bleeding of the gums, and bone loss (*Salvi et al., 2023*). The presence of periodontal pathogens and activation of the host immune response lead to the intensification of local inflammatory reactions and accumulation of metabolites (*Ji, Choi & Choi, 2015*). During the pathological process of periodontitis, lactic acid produced by bacterial metabolism causes acidification of the local environment, thereby affecting related physiological activities (*Ishikawa et al., 2021*). As the final product of glycolysis, the increased concentration of lactic acid in the periodontitis microenvironment results in elevated levels of lactylation. This change in modification significantly impacts cell function and inflammatory responses (*Chen et al., 2022b*).

In the periodontitis microenvironment, macrophages serve as crucial immune cells, and changes in their polarization status have a significant impact on inflammation progression and tissue repair (*Schlundt et al., 2021*). Intriguingly, lactylation has been demonstrated in numerous studies to regulate the polarization state and function of macrophages, thereby inhibiting inflammation progression and promoting tissue repair. Research shows that in a rat periodontitis model, after inhibiting the lactylation "writer" enzyme P300 in Raw264.7 cells, the lactylation level decreased due to the lack of a "starter", leading to increased expression of inflammatory cytokines IL-1β, IL-6, and TNF-α, which are key

**Table 1  Lactylation modification in bone metabolism and related diseases.** Systematic overview of lactylation-mediated regulation in various diseases, including affected cell types, downstream genes/proteins, and functional consequences.

| Category | Related items | Role/effects of lactylation modification |
|---|---|---|
| Diseases | Periodontitis | Lactylation modulates macrophage polarization (inhibits M1, promotes M2) and osteogenic differentiation of stem cells (*e.g.*, activates Wnt/β-catenin in PDLSCs), reducing inflammation and bone resorption. *Porphyromonas gingivalis* infection increases lactylation, promoting Aβ production and bone loss. |
| | Osteoporosis | Lactylation *via* endothelial cell-derived lactate activates H3K18la in BMSCs, upregulating osteogenic genes (*e.g.*, *COL1A2*, *COMP*). High-intensity interval training (HIIT) enhances bone anabolism through lactate accumulation. |
| | Cleft palate (CP) | Maternal *P. gingivalis* infection upregulates embryonic H4K12la, suppressing ADAM17/MerTK signaling and causing CP. |
| | Osteoarthritis (OA) | Lactylation of UGDH (K6 site) inhibits glycosaminoglycan synthesis and activates pro-inflammatory MAPK signaling. |
| | Rheumatoid arthritis (RA) | PKM2 lactylation in synovial fibroblasts inhibits nuclear translocation and proliferation signals, alleviating joint inflammation. |
| Cells | Osteoblasts/Stem cells | Lactylation (*e.g.*, H3K18la) activates osteogenic genes (*RUNX2*, *JunB*). P300 act as "writers" to enhance lactylation levels. |
| | Osteoclasts | Lactylation suppresses osteoclast markers (*TRAP*, *CTSK*) via P300/H3K18la/HDAC1. However, lactate-induced acidosis may promote osteoclast activity (mechanism unclear). |
| | Macrophages | Lactylation drives the polarization of M1 macrophages toward the M2 phenotype, potentially by promoting the secretion of anti-inflammatory cytokines (IL-4, IL-13) and osteogenic factors (BMP-2), thereby suppressing inflammation and facilitating bone repair. |
| | T Cells (Th17/Treg) | Lactylation suppresses Th17 differentiation through Ikzf1 lactylation while enhancing Treg function (*e.g.*, *via* strengthened TGF-β signaling), potentially modulating bone-immune homeostasis. In CD8+ T cells, lactylation may influence the bone microenvironment by activating effector genes (Gzmb, IFN-γ). |
| Genes | *RUNX2, JunB, COL1A2, COMP, ENPP1, TCF7L2* | Direct activation by lactylation (*e.g.*, H3K18la) promotes osteogenesis. |
| | *TRAP, CTSK, MMP9, ATP6V0D2* | Epigenetic suppression via lactylation reduces osteoclast activity. |
| | *Arg1, RARγ* | *Arg1* (M2 marker) is upregulated by lactylation, *RARγ* is downregulated in macrophages, potentially affecting osteogenesis. |
| Proteins/Pathways | p300 | p300 mediates histone lactylation (*e.g.*, H3K18la). Both act as "writers" for osteogenic differentiation. |
| | HDACs | HDAC1/3 exhibit histone lysine delactylase activity, capable of removing lactylation modifications (*e.g.*, H3K18la) and reversing lactylation-mediated gene regulation. |
| | UGDH, PKM2 | UGDH lactylation (K6) inhibits enzymatic activity, activating MAPK inflammation; PKM2 lactylation blocks nuclear translocation in RA. |
| | Wnt/β-catenin, P300/H3K18LA/HDAC1 | Lactylation activates Wnt signaling for osteogenesis, suppresses osteoclastic bone resorption activity by modulating the P300/H3K18la/HDAC1 pathway. |

pro-inflammatory factors in the inflammatory response. Simultaneously, the levels of M1 macrophage-associated factors such as CD86 and iNOS increased, while those of M2-associated factors Arg1 and CD206 decreased (*Liu et al., 2024b*). This experiment directly reveals a possible mechanism of lactylation in the development of periodontitis, providing a more comprehensive understanding of the disease. In another study, transfection of macrophages with a novel ncRNA from *Porphyromonas gingivalis*, namely msRNA P.G_45033, increased lactate and lactylation levels in macrophages, inducing an increase in β-amyloid protein (Aβ) production (*Zhang et al., 2023*). Elevated Aβ levels have a significant impact on bone loss (*Li et al., 2014*; *LLabre et al., 2020*). These studies link periodontitis, macrophages, and lactylation, uncovering a novel mechanism of lactylation's role in alveolar bone resorption in periodontitis.

Furthermore, lactylation can influence bone resorption and reconstruction processes in periodontal tissues by regulating osteogenic differentiation of stem cells. For instance, in periodontitis, proanthocyanidins can ameliorate the inhibition of osteogenic differentiation in PDLSCs under lipopolysaccharide (LPS)-induced inflammatory conditions. Specifically, proanthocyanidins achieve this by restoring lactylation in PDLSCs, upregulating the Wnt/β-catenin pathway, and increasing the expression of the downstream target gene RUNX2, thereby promoting alveolar bone regeneration (*Wu & Gong, 2024*). Similarly, scopolamine (SCO) treatment of PDLSCs also favors periodontal tissue regeneration, with a process akin to the aforementioned experiment with proanthocyanidins (*Wu & Gong, 2024*). Additionally, histone lactylation, by mediating lactate levels, can regulate alveolar bone remodeling during orthodontic tooth movement (*Zhai et al., 2022*). These studies highlight the significance of lactylation in the development and improvement of periodontitis, providing a more comprehensive understanding of the pathogenesis of periodontitis and potential strategies for targeted therapy.

However, these studies also have limitations. First, most research is based on cell or animal models (such as Raw264.7 cells), with insufficient direct evidence from human periodontal tissue. Second, there is limited research on the direct effects of lactylation on osteoclasts (as opposed to osteoblasts), and the bidirectional regulation of bone resorption and formation has not been fully confirmed. Finally, high lactate environments may exacerbate inflammation (*Arnett & Orriss, 2018*; *Yuan et al., 2016*), suggesting that lactylation could have opposing effects due to differences in the microenvironment, necessitating more detailed spatiotemporal dynamic studies.

## Lactylation and osteoporosis

Osteoporosis is a prevalent bone disease primarily characterized by decreased bone mass and deterioration of bone microarchitecture, leading to increased bone fragility and susceptibility to fractures. As individuals age, bone metabolism gradually slows down, resulting in progressive bone mass loss and an elevated risk of osteoporosis. This risk further intensifies, particularly in women after menopause (*Lorentzon et al., 2022*; *Sànchez-Riera et al., 2010*). Additionally, several metabolic syndromes, including central obesity, hyperglycemia, hypertension, and dyslipidemia, have been associated with osteoporosis (*Wong et al., 2016*).

Lactate can act as a signaling molecule to influence bone metabolism. Studies have shown that lactate, beyond serving as an energy source, also functions as a signaling molecule impacting various cellular functions. In the development of osteoporosis, lactate may regulate bone mineral density by affecting the activity of osteoblasts and osteoclasts (*Taubmann et al., 2020*; *Wu et al., 2017*). Intriguingly, lactate can influence bone cell function through histone lactylation. Research has found that endothelial cells (ECs) produce lactate during glycolysis and permeate it into bone tissue through blood vessel walls, triggering H3K18la in bone marrow mesenchyml stem cells (BMSCs), thereby activating the expression of bone formation-related genes. Specific mechanisms include lactate-induced expression of bone formation-related genes such as COL1A2, COMP, ENPP1, and TCF7L2 in BMSCs, which play crucial roles in bone mineralization and formation. Conversely, when ECs glycolysis are inhibited, serum lactate levels decrease, along with reduced H3K18la in BMSCs, negatively impacting bone formation. By restoring lactate levels, such as through exogenous lactate supplementation or high-intensity interval exercise, osteoporotic symptoms can be partially reversed. Furthermore, clinical data also indicate that osteoporotic patients exhibit lower serum lactate levels, decreased H3K18la levels in BMSCs, and reduced expression of related genes, further supporting the important role of lactylation in bone formation. Regarding bone vascular density, mice exhibit decreased bone vascular density and reduced expression of PKM2, a regulatory factor of endothelial glycolysis. This suggests that the glycolysis process in endothelial cells has a significant impact on bone vascular density, and lactate, as one of the products of glycolysis, may be involved in this process (*Wu et al., 2023a*).

In terms of improving osteoporosis, high-intensity interval training (HIIT) has been shown to enhance bone mineral density (BMD) in osteoporotic animals through lactate-mediated anabolic bone effects, suggesting that lactate produced during exercise may promote bone health by enhancing osteoblast activity and mineralization capacity through promoting protein lactylation (*Huang et al., 2023*; *Zhu et al., 2023*). In addition, hypertension, which is also a common condition among the elderly, is closely related to osteoporosis. Possible factors contributing to this relationship may include nutrient intake, cytokine release, or signaling pathway activation (*Do Carmo & Harrison, 2020*; *Nakagami & Morishita, 2013*; *Varenna et al., 2013*). Exercise increases lactate production and induces NICD lactylation, thereby inhibiting NOTCH signaling, upregulating the expression of angiogenic factors, promoting angiogenesis, and lowering blood pressure (*Liu et al., 2024c*).

Overall, lactylation plays a significant role in regulating osteoporosis. Through various mechanisms such as promoting osteoblast differentiation, regulating bone vascular density, and facilitating bone formation, lactylation contributes to maintaining bone health and homeostasis. Furthermore, through exercise, lactate accumulation in the body can enhance lactylation, which may directly influence bone cell activity, offering a potential means of alleviating osteoporosis. From an indirect perspective, lactylation can also affect diseases associated with osteoporosis, such as improving hypertension, which in turn may help restore bone health (*Ilić, Obradović & Vujasinović-Stupar, 2013*). In conclusion, these studies demonstrate the link between lactylation, osteoporosis, and exercise, positioning lactylation as a new "mediator" for osteoporosis treatment.

## OTHER DISEASES

In the oral cavity, lactylation is closely associated with the development of the jawbone. *Porphyromonas gingivalis* (*P. gingivalis*), a pathogen strongly linked to periodontal disease, has previously been suggested to potentially affect fetal development through maternal infection (*Michelin et al., 2012*). *Zhao et al. (2025)* exposed pregnant mice to *P. gingivalis* and demonstrated that *P. gingivalis* remodels the metabolic and epigenetic state of the embryonic palatal microenvironment, resulting in enhanced glycolysis, upregulated histone lactylation (H4K12la), and the subsequent inhibition of the ADAM17-mediated MerTK/TGFBR1 signaling pathway, ultimately leading to cleft palate. This finding provides a new molecular perspective for understanding the connection between maternal periodontal disease, lactylation modification, and congenital cleft palate in offspring.

Moreover, lactylation plays a significant regulatory role in osteoarthritis (OA) and rheumatoid arthritis (RA). A study by *Lan et al. (2025)* found that the lactate levels in the synovial fluid and in chondrocytes under inflammatory stimulation were significantly elevated in OA patients, leading to an increase in protein lactylation. Notably, lactylation at the K6 site of UGDH was identified as a central regulatory event. This lactylation modification not only inhibited the enzymatic activity of UGDH, reducing glycosaminoglycan synthesis, but also promoted the translocation of UGDH from the nucleus to the cytoplasm. This translocation weakened its binding capacity with STAT1, thereby relieving the suppression of MAP3K8 transcription and ultimately activating the pro-inflammatory and catabolic MAPK signaling pathway (*Lan et al., 2025*). This study revealed that UGDH K6 lactylation promotes OA inflammation and cartilage degradation by inhibiting glycosaminoglycan synthesis and activating the MAPK pathway, highlighting the interplay between the metabolic-epigenetic-inflammation axis.

During the progression of RA, the glycolytic metabolism of fibroblast-like synoviocytes (FLS) is enhanced, producing large amounts of lactate and inducing lactylation on PKM2 protein. Studies have shown that the natural drug artemisinin (ART) can regulate this lactylation by directly interacting with PKM2. ART promotes the binding of PKM2 to the acetyltransferase p300, increasing the degree of PKM2 lactylation, thereby inhibiting the nuclear translocation of PKM2, blocking its proliferative signaling in FLS, and inducing cell cycle arrest. In animal experiments, ART treatment significantly reduced synovial hyperplasia and joint damage in RA, and alleviated arthritis-related pain and inflammatory responses (*Wang et al., 2024b*). In summary, lactylation regulates the proliferation and migration of FLS in RA by affecting the function of PKM2, providing a potential new direction for RA treatment (Table 1).

As a crucial aspect in the pathogenesis of diseases such as periodontitis and osteoporosis, research on lactylation may offer new therapeutic targets for these diseases, for instance, by developing drugs that can reverse or regulate lactylation (*Moreno-Yruela et al., 2022*). By inhibiting or promoting lactylation, the progression and prognosis of diseases can be regulated. Currently, drug development targeting lactylation primarily focuses on small molecule inhibitors of MCT1 and LDH inhibitors, which can affect lactate transport and production, thereby regulating the level of lactylation (*Chen et al., 2022b*).

## CONCLUSION

Lactylation, as an emerging post-translational modification, has been found to play pivotal roles in various cell types in recent years, particularly in metabolically active cells. In the complex physiological process of bone metabolism, lactylation is increasingly recognized as potentially involved in regulating the functions of osteoblasts, influencing the metabolic activities and gene expression of these cells (*Sasa, Kato & Yamada, 2024*). Osteoblasts are the cell type responsible for new bone formation and have extremely high metabolic demands, especially during bone formation (*Lee et al., 2017*). Lactylation modification may promote bone matrix synthesis and mineralization by regulating the metabolic pathways of osteoblasts, particularly the balance between glycolysis and oxidative phosphorylation. For instance, lactate, as the end product of glycolysis, may modify histones through lactylation, thereby regulating the expression of osteoblast-related genes and affecting the differentiation and function of osteoblasts (*Nian et al., 2022*; *Wu et al., 2023a*; *Wu et al., 2024*). Osteoclasts play a crucial role in bone resorption, and their activity depends on energy metabolism processes, especially their ability to degrade bone matrix in an acidic environment. Studies have shown that the accumulation of metabolites such as lactate in osteoclasts can affect the function and differentiation of these cells (*Arnett & Orriss, 2018*; *Yuan et al., 2016*). However, the specific molecular regulatory mechanisms remain unclear. Lactylation may influence the bone resorption function of osteoclasts by regulating the activity of metabolic enzymes within them, which warrants further exploration. This metabolic regulation is particularly significant in metabolic bone diseases such as osteoporosis, as excessive osteoclast activity is closely related to excessive bone resorption (*Novack & Teitelbaum, 2008*).

The role of lactylation in bone metabolism suggests its potential as a target for regulating bone metabolism-related diseases such as periodontitis, osteoporosis, and osteoarthritis (*Liu et al., 2024b*; *Wang et al., 2024b*; *Wu et al., 2023a*). By modulating the level of lactylation, it may help restore the balance of bone metabolism, thereby enabling the development of new therapeutic strategies. For example, promoting lactylation in osteoblasts can contribute to increased bone formation (*Wu et al., 2023a*).

Although the specific molecular mechanisms of lactylation in bone metabolism have not been fully elucidated, it is increasingly being recognized as a key node in metabolic regulation. Future research may focus on how to leverage the regulatory mechanisms of lactylation to develop new treatment options for bone diseases (*Liu et al., 2024b*), as well as gaining a deeper understanding of how lactylation synergizes with other metabolic pathways, such as acetylation and phosphorylation, to maintain the balance of bone metabolism.

## LIMITATIONS, CHALLENGES, AND PERSPECTIVES

### Histone lactylation and metabolism: which comes first, the chicken or the egg?

Metabolism influences post-translational modifications of histone proteins by regulating the availability of donor substrates, whereas histone modifications, in turn, can impact

metabolic pathways by modulating the expression of key metabolic enzymes and altering metabolite pools. Although the detailed crosstalk between histone lactylation and cellular metabolism falls outside the scope of this review, it is important to note that metabolic reprogramming alone can significantly affect stem cell differentiation. For a more comprehensive discussion on the interplay between metabolism and cell fate decisions, readers are referred to the review by *Wu, Ocampo & Belmonte (2016)*

However, emerging evidence increasingly supports the notion that histone lactylation acts as a primary regulatory mechanism in bone cell differentiation, rather than being a mere byproduct of metabolic changes. For example, in a study by *Minami et al. (2023)* silencing the histone lactylation enzyme p300 significantly inhibited osteoblast differentiation. This was accompanied by a marked reduction in histone lactylation levels, suggesting that p300-mediated lactylation is essential for the osteogenic process. In a more recent and mechanistically thorough study, (*Wu et al., 2023a*) demonstrated that endothelial cell-derived lactate promotes osteogenesis in BMSCs through histone lactylation. Knockdown of MCT1, the monocarboxylate transporter responsible for lactate uptake, reduced intracellular lactate levels, suppressed H3K18la modification, and significantly decreased ALP and ARS staining in BMSCs treated with conditioned medium from bone microvascular endothelial cells (BMECs). Furthermore, genetic deletion or pharmacological inhibition of p300 led to reduced histone lactylation and impaired osteogenic differentiation under the same conditions. Together, these studies provide compelling functional evidence that lactylation is not merely reflective of metabolic flux but plays a direct and causative role in regulating bone cell fate.

Although current evidence directly linking histone lactylation to osteoblast metabolism and differentiation remains limited, owing to the relative novelty of the field, emerging insights from related epigenetic modifications suggest a plausible regulatory axis. For instance, in calcific aortic valve disease, mechanical stress sensed through the Piezo1 ion channel promotes glutaminolysis and acetyl-CoA production, enhancing H3K27 acetylation at osteogenic genes mediated by RUNX2 in valve interstitial cells (*Zhong et al., 2023*). Similarly, hypoxia-induced citrate export from mitochondria to the cytoplasm increases cytosolic acetyl-CoA, supporting H3K27 acetylation and osteogenic gene expression in mesenchymal stem cells (*Pouikli et al., 2022*). These examples highlight how metabolic shifts can regulate histone modifications to control cell fate decisions. However, whether histone lactylation follows a similar paradigm in osteoblasts or osteoclasts remains an open question that future studies will need to address.

### Limitations

To date, no *in vivo* models have directly and specifically demonstrated the causal role of histone lactylation in bone formation or resorption. While lactylation is a downstream product of lactate metabolism, its functional dissection *in vivo* remains challenging due to multiple overlapping mechanisms and confounding metabolic effects.

Strategies to inhibit histone lactylation typically involve targeting lactylation "writers" or "erasers" through genetic or pharmacological approaches, or modulating intracellular lactate availability. However, global knockout of lactylation "writers" CBP/p300 is

embryonically lethal, limiting the use of conventional knockout models (*Kasper et al., 2006*). A recent study demonstrated that conditional deletion of p300 in osteoblasts leads to significantly reduced bone mass and mechanical strength, highlighting its essential role in bone homeostasis (*Zhang et al., 2024*). Nevertheless, as p300 mediates not only lactylation but also histone acetylation and other acylations, its deletion results in widespread epigenetic alterations, thereby confounding efforts to attribute phenotypic effects specifically to the loss of histone lactylation. Moreover, p300 has been shown to promote osteogenesis through transcriptional coactivation of signaling pathways such as SMAD1 and Osterix, further complicating its interpretation as a lactylation-specific effector in bone biology (*Quan et al., 2023*).

Inhibiting lactate transport (*e.g.*, *via* MCT1) or synthesis (LDHA) represents another approach, though these strategies also have limitations. MCT1 blockade may impair mitochondrial biogenesis and metabolic flux, thereby indirectly altering cell function (*Zhang et al., 2024*), and it cannot prevent intracellular lactate production. Moreover, osteoblasts rely primarily on glycolysis for energy, while osteoclasts depend more on oxidative phosphorylation (*Lee et al., 2017*), suggesting that LDHA inhibition may offer a more targeted model in osteoblasts. In vitro studies have shown that LDHA knockdown impairs ALP activity and mineralized nodule formation, supporting a role for lactylation in osteoblast differentiation (*Nian et al., 2022*). However, a bone cell-specific LDHA knockout model *in vivo* has not yet been reported, due to the broad role of LDHA in cellular energy metabolism and the risk of disrupting systemic physiological processes (*Kwarteng & Agathocleous, 2023*). It is also important to recognize that bone cells are influenced by extracellular lactate originating from non-bone cells such as endothelial cells and skeletal muscle cells. These exogenous lactate sources can be taken up *via* MCTs, or extracellular vesicles (EVs) and drive histone lactylation in osteoblasts or BMSCs (*Wu et al., 2023a*). Therefore, osteoblast- or osteoclast-specific LDHA deletion may not eliminate functional lactylation.

Furthermore, lactate itself can activate signaling pathways independent of histone modification. For instance, it binds to GPR81, a lactate-sensing receptor on osteoblasts, and enhances osteogenic activity through PKC-Akt signaling, independently of lactylation (*Wu et al., 2018*). Hence, *in vivo* suppression of lactate production or transport introduces multiple metabolic and signaling confounders that limit our ability to attribute observed effects solely to histone lactylation.

Nevertheless, indirect evidence supports the physiological relevance of histone lactylation in bone. *Wu et al. (2023a)* demonstrated that endothelial cell-specific knockout of PKM2, a key glycolytic regulator, reduced lactate production, which in turn decreased H3K18la levels in BMSCs and impaired osteogenesis. They further identified COL1A2, COMP, ENPP1, and TCF7L2 as downstream targets of H3K18la *via* GO enrichment and transcriptome analysis. Rescue experiments using exogenous lactate and PKM2 overexpression confirmed that endothelial cell-derived lactate promoted osteogenesis *via* histone lactylation.

Although these findings remain indirect, the experimental strategy is consistent with approaches used in multiple studies across different systems to investigate lactylation,

including myocardial infarction, sepsis, and cancer models (*Hao et al., 2024*; *Wang et al., 2022b*; *Yang et al., 2022*; *Zhang et al., 2019*).

## Perspectives

Lactylation, along with other post-translational modifications such as acetylation, phosphorylation, and methylation, may collectively participate in complex gene regulatory networks (*Chen et al., 2022b*; *Liu et al., 2022*). Future research directions should include exploring the synergistic or antagonistic effects of lactylation with these modifications, especially in regulating chromatin states, gene expression, and cell fate determination. By integrating the interactions among multiple post-translational modifications, it may be possible to uncover more complex cellular metabolic and signaling pathways (*Cheng et al., 2023*; *Wu et al., 2023b*).

With the increasing incidence of metabolic diseases such as diabetes and obesity, the potential role of lactylation in these diseases has garnered significant attention (*Wu et al., 2023b*). Future research may focus on elucidating how lactylation participates in regulating metabolic imbalances, particularly its pivotal role in the metabolic reprogramming of osteoblasts (*Sasa, Kato & Yamada, 2024*). By modulating the levels of lactylation modifications, it may be possible to restore metabolic balance, thereby slowing down or reversing the progression of metabolic diseases. Furthermore, the regulation of lactylation may also be applied to enhance therapeutic effects on insulin resistance (*Maschari et al., 2022*). Additionally, the regulatory role of lactylation in inflammatory responses provides potential intervention pathways for the treatment of chronic inflammatory diseases. By inhibiting excessive inflammatory responses through the modulation of lactylation, lactylation modifications may be used in the treatment of inflammatory diseases such as rheumatoid arthritis (*Wang et al., 2024b*).

In summary, as an emerging post-translational modification, lactylation demonstrates its significant role in metabolic regulation, gene expression, and disease progression. With the further elucidation of its regulatory mechanisms and biological functions, its potential for clinical application will continue to increase, especially in the fields of bone regeneration, immunology, and metabolic disease treatment. As technological advancements progress, future research is expected to uncover the broader impacts of lactylation in cellular physiology and pathology, and provide new insights for disease treatment (*Gong et al., 2024*; *Hu et al., 2024*).

### Funding

This work was supported by the funding from the Guangzhou Science and Technology Program (2024A03J0140 and 2024A03J0064), Guangzhou Medical University 2024 Research Capacity Enhancement Program Major Clinical Research Projects (GMUCR2024-02023). The funders had no role in study design, data collection and analysis, decision to publish, or preparation of the manuscript.

## Grant Disclosures

The following grant information was disclosed by the authors:

Guangzhou Science and Technology Program: 2024A03J0140, 2024A03J0064.

Guangzhou Medical University 2024 Research Capacity Enhancement Program Major Clinical Research Projects: GMUCR2024-02023.

## Competing Interests

The authors declare there are no interest in competing.

## Author Contributions

- Zhiyuan Ye conceived and designed the experiments, performed the experiments, prepared figures and/or tables, authored or reviewed drafts of the article, and approved the final draft.
- Xuanyu Chen performed the experiments, analyzed the data, authored or reviewed drafts of the article, and approved the final draft.
- Jiyuan Zou performed the experiments, analyzed the data, authored or reviewed drafts of the article, and approved the final draft.
- Wenjing Wu analyzed the data, prepared figures and/or tables, authored or reviewed drafts of the article, and approved the final draft.
- Jingyuan Yang analyzed the data, prepared figures and/or tables, authored or reviewed drafts of the article, and approved the final draft.
- Li Yang conceived and designed the experiments, authored or reviewed drafts of the article, and approved the final draft.
- Lvhua Guo conceived and designed the experiments, authored or reviewed drafts of the article, and approved the final draft.
- Tao Luo conceived and designed the experiments, authored or reviewed drafts of the article, and approved the final draft.

## Data Availability

This is a literature review.

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
