# Peer review of "Lactylation’s role in bone health and disease: mechanistic insights and therapeutic potential"

_PeerJ, doi:10.7717/peerj.19534_

## Round 0.1 · original submission · Major Revisions

Please address the concerns of the reviewers and revise the manuscript accordingly.

Reviewer 1 ·

Basic reporting

Add related mechanistic diagrams to increase readability.
Please rewrite the “Other Diseases” section of “Lactation in Bone-Related Diseases” closely adhering to the title.

Experimental design

Some of the conclusions presented in the paper need to be supported by more documentation. (See the yellow highlighted section in the PDF for details)

Validity of the findings

No comments

Annotated reviews are not available for download in order to protect the identity of reviewers who chose to remain anonymous.

Reviewer 2 ·

Basic reporting

Lactylation's Role in Bone Health and Disease: Mechanistic Insights and Therapeutic Potential
This manuscript presents a comprehensive review of the emerging role of lactylation, a PTM, in bone metabolism and related diseases. The authors successfully consolidate existing knowledge on lactylation, highlighting its potential implications in osteoblast differentiation, osteoclast activity, immune cell function, and metabolic bone diseases. However, there are some suggestions that the authors should consider.
Comments:
1. The manuscript compiles evidence supporting lactylation’s role in bone metabolism but does not critically assess limitations of current studies. For example, how robust are the existing studies in establishing a causal role for lactylation in osteoblast differentiation or osteoclast activity? A discussion of potential metabolic shifts and histone acetylation crosstalks would strengthen the review.
2. The authors should address whether lactylation is a primary driver of bone cell differentiation or merely a secondary effect of metabolic alterations.
3. Although this review discusses lactylation is mediated histone acetyltransferases (P300/EP300), it does not discuss other specific erasers.
4. Are there bone-specific regulators of lactylation? What are the potential upstream signaling pathways that govern this modification in osteoblasts or osteoclasts?
5. Have any in vivo models demonstrated a direct impact of lactylation on bone formation or resorption? Please discuss.
6. This review could benefit from an elaborate table summarizing the review. Also, Biorender/or other Illustrations could make this review look better and will benefit a larger audience.

Experimental design

The review could benefit from a comprehensive table and at least one Illustration.

Validity of the findings

No comment

Additional comments

No comments

---

## Round 0.2 · accepted · Accept

All issues pointed by the reviewers were accurately addressed and revised manuscript is acceptable now.